# Accelerated Inference and Reduced Forgetting: The Dual Benefits of Early-Exit Networks in Continual Learning

## Abstract

In the pursuit of a sustainable future for machine learning, energy-efficient neural network models are crucial. A practical approach to achieving this efficiency is through early-exit strategies. These strategies allow for swift predictions by making decisions early in the network, thereby conserving computation time and resources. However, so far the early-exit neural networks have only been developed for stationary data distributions, which restricts their application in real-world scenarios where training data is derived from continuous non-stationary data. In this study, we aim to explore the continual training for early-exit networks. Specifically, we adapt the existing continual learning methods to fit early-exit architectures and introduce task-aware dynamic inference to improve the network accuracy for a given compute budgets. Finally, we evaluate continually those methods on the standard benchmarks to assess their accuracy and efficiency. Our work highlights the practical advantages of using the early-exit networks in more real-world continual learning scenarios.

## 1 Introduction

Driven by the demand for energy-efficient employment of deep neural networks, early-exit methods have experienced a notable increase in research attention (Scardapane et al., 2020b; Matsubara et al., 2022). These methods reduce computation for easier samples that do not require the full computational complexity of the applied network. Early-exit methods introduce internal classifiers (ICs) at the intermediate layers of the network and allow the network to return the prediction of the first IC sufficiently confident about its prediction. ICs leverage the fact that some samples are easier to classify, and therefore can be processed by a fraction of the network, reducing the compute spent on the processing. Interestingly, this technique was found not only to reduce computation but could also improve accuracy by preventing *overthinking* (Kaya et al., 2019) on samples that can be confidently classified by early ICs. Existing early-exit methods have been developed for training on i.i.d data scenarios, where all training samples are jointly available. However, for many real-world applications, such joint training is unrealistic and algorithms have to be continually trained from a sequences of data (also called tasks), without (or with only limited) access to data from previous tasks. Therefore, in this paper, we aim to investigate the continual training of early-exit networks.

The field of continual learning proposes theory and algorithms for continual learning on a non-i.i.d stream of data (De Lange et al., 2021). The main challenge for continual learning is called *catastrophic forgetting* and refers to a significant drop in the learner's performance on data from previous tasks (McCloskey & Cohen, 1989; Kirkpatrick et al., 2017). Several main strategies have been developed to address this problem. Among them, parameter isolation (Rusu et al., 2016; Serra et al., 2018; Mallya & Lazebnik, 2018), weight and data regularization (Aljundi et al., 2018; Kirkpatrick et al., 2017; Li & Hoiem, 2017) and rehearsal methods (Rebuffi et al., 2017; Chaudhry et al., 2018) are the most popular. In this paper, we investigate continual training of regularization and rehearsal methods of early-exit networks with a focus on the more challenging class-incremental (Van de Ven & Tolias, 2019) setting.

We start our work with an in-depth analysis of the continual learning of early-exit networks implemented on several standard continual learning methods. Interestingly, we observe that catastrophic

forgetting is not uniformly distributed along the network layers, but is more prominent in the later layers of the network. Instead, early internal classifiers suffer less from catastrophic forgetting and become relatively more confident with respect to the later (and more complex) internal classifiers. Furthermore, we show that *overthinking* is more detrimental for continually learned networks than for jointly trained networks. As a consequence, continually trained early-exit networks have potentially a better trade-off between inference-time computation and performance. Finally, we notice that dynamic inference in early-exit networks is negatively affected by task-recency bias and propose a simple method of task-aware dynamic inference to improve the accuracy of early-exit network inference.

The main contributions of our work are:

- We are the first work to investigate the continual learning of early-exit networks and perform a comprehensive evaluation of various continual learning methods enhanced with early-exit capabilities.

- Interestingly, our analysis shows that early-exit classifiers do not only reduce inference time, but they can also counter the forgetting of previously seen data. In addition, we show that, with respect to the conventionally trained networks, *overthinking* is a more prominent problem in continually trained ones.

- We show that the task-recency bias negatively affects dynamic inference in early exit networks and propose a simple method of task-aware thresholding to improve the compute-accuracy trade-off.

- Early-exit networks can improve performance for several continual learning methods. Also, they can be applied to obtain inference time speed-ups from 25 to 60% with equal accuracy compared to the method without early-exit (see Figure 4). Notably, for the popular exemplar-free method LwF, we can obtain a significant performance gain (+5%) or a speed-up of around 50% with equal performance to standard LwF.

## 2 RELATED WORK

### 2.1 ADAPTIVE INFERENCE IN DEEP NEURAL NETWORKS

Many methods have been crafted to reduce the computational cost of training and inference of deep neural networks. Several network architectures were designed with efficiency in mind (Tan & Le, 2019), and methods such as knowledge distillation (Hinton et al., 2015), pruning or quantization (Liang et al., 2021) aim to reduce the complexity of the model during the inference. Additionally, multiple works also try to reduce resource usage through the introduction of conditional computations. Methods like Mixture-of-Experts (Shazeer et al., 2017) adapt the computational path of the model to an input sample. Some works adapt the compute by selecting the subset of filters (Liu et al., 2019; Herrmann et al., 2020), features (Figurnov et al., 2017; Verelst & Tuytelaars, 2020) or tokens (Riquelme et al., 2021) processed in each layer. Works such as (Lin et al., 2017; Nie et al., 2021) introduce sparsity while training the model. Multiple methods Graves (2016); Wang et al. (2018); Dehghani et al. (2018); Elbayad et al. (2019); Banino et al. (2021) allow the model to adapt its depth to the input example via skipping layers (Wang et al., 2018), competing halting scores (Graves, 2016; Banino et al., 2021), introducing recursive computations (Dehghani et al., 2018; Elbayad et al., 2019) or attaching early exits (Scardapane et al., 2020b; Matsubara et al., 2022). Aside from early exits, most of those methods require a specific model architecture or training scheme and are not easily applicable to practical use cases. Therefore, in this work, we focus on the early-exit framework, as it is easy to apply to most modern neural networks and straightforward when combined with other techniques such as pruning, quantization or knowledge distillation.

### 2.2 EARLY-EXIT NETWORKS

Early exits (EE) allow for dynamic inference by adding internal classifiers (ICs) at the various layers of the network and returning the prediction from the first IC that satisfied the exit criterion. Panda et al. (2016) was one of the first to use linear classifiers as ICs for EE. Teerapittayanon et al. (2016) proposed to use the entropy of IC prediction as the exit criterion. Berestizshevsky & Even (2019)

proposed to use prediction confidence as the exit criterion, and Kaya et al. (2019) suggested selecting the placement of ICs based on the computational cost of the network layers. Zhou et al. (2020) proposed to decide on the exit based on the alignment of predictions from previous ICs. Several works proposed to improve ICs in EE models by adding cascading connections (Li et al., 2019), ensembling IC predictions (Sun et al., 2021) or combining both (Wołczyk et al., 2021; Liao et al., 2021). Other works, such as (Scardapane et al., 2020a; Han et al., 2022; Yu et al., 2023), proposed training schemes designed for EE models. EE methods also received significant attention in NLP (Zhou et al., 2020; Xin et al., 2020; Schwartz et al., 2020) as they fit well with the Transformer architecture.

### 2.3 CONTINUAL LEARNING

Continual learning methodologies (Parisi et al., 2019; De Lange et al., 2021; Masana et al., 2022) can be broadly classified into three categories: *regularization-based*, *replay-based*, and *parameter-isolation* methods. *Regularization-based* approaches typically introduce a regularization term in the loss function to constrain changes to parameters relevant to prior tasks. These can further be categorized as data-focused (Li & Hoiem, 2017), leveraging knowledge distillation from previously trained models, or prior-focused (Kirkpatrick et al., 2017; Zenke et al., 2017; Aljundi et al., 2018), estimating parameter importance as a prior for the new model. Recent research proposed enforcing weight updates within the null space of feature covariance (Wang et al., 2021; Tang et al., 2021). *Replay-based* methods rely on memory and rehearsal mechanisms to recall episodic memories of past tasks during training, thereby keeping the loss low in those tasks. Two main strategies are: exemplar replay - which stores selected training samples (Riemer et al., 2018; Buzzega et al., 2020; Chaudhry et al., 2018; 2019) and generative replay - with models that synthesize previous data with generative models (Shin et al., 2017; Wu et al., 2018). *Parameter isolation* methods aim to learn task-specific sub-networks within a shared network. Various techniques, such as Piggyback (Mallya et al., 2018), PackNet (Mallya & Lazebnik, 2018), SupSup (Wortsman et al., 2020), HAT (Serra et al., 2018), and Progressive Neural Network (Rusu et al., 2016), allocate and combine parameters for individual tasks. While effective in task-aware settings, these methods are most suited for scenarios with a known task sequence or oracle.

## 3 CONTINUALLY LEARNING EARLY-EXIT MODELS

### 3.1 PROBLEM SETUP

We consider a supervised continual learning (CL) scenario, where a learner (neural network) is trained over $T$ classification tasks and its goal is to learn to classify the new classes while avoiding catastrophic forgetting of the previously learned ones. At each task $t$, the model can only access the dataset $\mathcal{D}_t = \{\mathcal{X}_t, \mathcal{Y}_t\}$, which is composed of a set of input images $\mathcal{X}_t$ and corresponding labels $\mathcal{Y}_t$. Our study focuses on the offline setting, where the learner can pass through the data samples from the current task multiple times.

We examine two standard CL settings: task-incremental learning (TIL) and class-incremental learning (CIL). In both TIL and CIL scenarios, each task data $D_t$ consists of a disjoint subset of classes. The key distinction between CIL and TIL lies in the assumption that in CIL the learner is unable to access the task-id during inference. Therefore, in CIL the learner needs to be able to learn to distinguish between all the classes encountered so far. In contrast, in TIL it knows the task identity during the inference time and only needs to choose from the smaller subset of classes from that particular task. In our study, we refer to these settings as task-agnostic (TAG) and task-aware (TAW) continual learning respectively.

### 3.2 EARLY-EXIT MODEL IN CONTINUAL LEARNING SETTING

Contrary to the classic deep neural networks commonly studied in the continual learning scenario, we focus on learning dynamic networks, specifically early-exit models. Early-exit model $\theta$ extends the standard neural network with $M$ internal classifiers (ICs) attached at the intermediate layers of the network. The network classifies its inputs incrementally and at each IC, if the classifier is sufficiently confident, it can return its prediction and skip the rest of the computations. If the

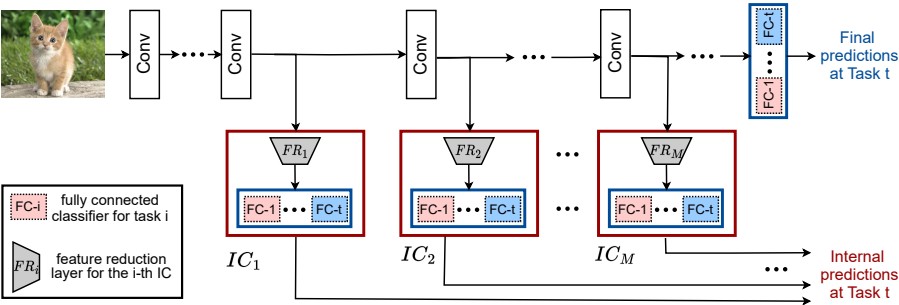

Figure 1: **Early-exit network in the continual learning setting after $t$ tasks.** At the different stages of the network, we attach internal classifiers (ICs) composed of the feature reduction (FR) layer and a multi-head class-incremental classifier with the fully-connected (FC) heads added over the course of the learning.

network does not exit at any of the ICs, it returns the prediction of the final classifier. ICs employ feature reduction (FR) layers to reduce the size of the inputs for fully connected networks (FC) that produce the predictions. The final classifier does not require the FR layer, as the network performs the feature reduction before this classifier by design. All the classifiers in our model are composed of multiple heads, and we add a new head upon encountering a new task. We depict the early-exit model in the CL setting in Figure 1.

We follow Shallow-Deep Neural Network (SDN) Kaya et al. (2019) architecture and place the ICs at the layers that require roughly 15%, 30%, 45%, 60%, 75%, 90% of the computations of the full network. During the inference, network is allowed to exit if the confidence of the IC prediction exceeds the threshold $\tau$. The confidence for each class is computed as the softmax over the network outputs.

For continual learning of early-exit neural network, the primary objective function at each training step $t$ in both TAW and TAG continual learning scenarios is the same and can be described as:

$$\arg\min_\theta \mathcal{L}_f\left(C^t(E(\mathcal{X}_t; \theta)), \mathcal{Y}_t\right) + \sum_{i=1}^{M} w_i * \mathcal{L}_i\left(IC_i^t(E_i(\mathcal{X}_t; \theta_i)), \mathcal{Y}_t\right), \tag{1}$$

where $L_i$ is the loss function for $i$-th IC, $w_i$ represents the weight assigned to the loss for this IC, and $E_i$ stands for the feature vector at the level of $IC_i$ produced by a the part of network with $\theta_i$ parameters. We denote the feature vector of the full network as $E$, and the final classifier as $C$. We jointly train the ICs and the final classifier, updating all the parameters $\theta$ of the network. Different loss weights for each IC serve to stabilize the training and mitigate overfitting in the earlier layers, which may have lower learning capacity.

### 3.3 Adapting continual learning methods to early exit networks

Most of the methods dedicated to the continual learning setup have been developed only for the networks with a single classification layer at the end, but the introduction of the additional ICs significantly affects the network learning dynamics. Therefore, we need to adapt those methods for use with the early-exit network. In most cases, we replicate the logic from the original method to all ICs. We list all adapted method below, with additional information whenever more decisions about adaptation were required.

For the exemplar-free setting, we evaluate **Learning without Forgetting (LwF)** (Li & Hoiem, 2017), where we maintain the same strength of distillation for all classifiers. For reference, we also evaluate the results for standard **Finetuning (FT)**.

For the setting with memory, we implement **Experience Replay (ER)** (Riemer et al., 2018) using balanced batches of data sampled from memory and from the new task. We also consider **Finetuning with Exemplars (FT-E)** (Belouadah & Popescu, 2019; Masana et al., 2022), where samples from memory and a current task are mixed before a training phase. We also adapt three classic continual learning strategies. We implement **Bias Correction (BiC)** (Wu et al., 2019) with an additional

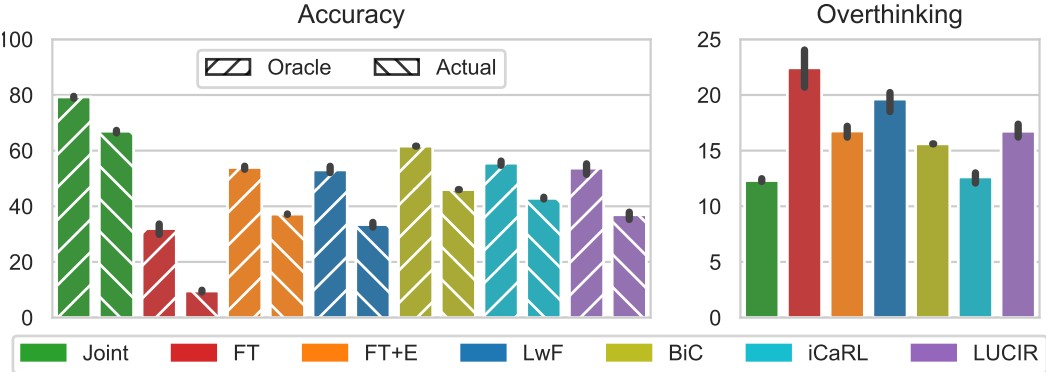

Figure 2: Analysis of overthinking in continually trained early-exit networks. (left) Ideal accuracy of early exit networks and the actual accuracy of the final classifier for continual learning methods. (right) *overthinking* defined as the difference between the oracle and the actual accuracy. Observe that overthinking is more prominent in continual learning scenarios, indicating that the potential gain from using early-exit models in this setting might be higher.

prediction rectification layer and knowledge distillation is added for each IC. In **LUCIR** (Hou et al., 2019), we make each IC a cosine classifier and apply feature balancing to the ICs separately. Finally, for **iCaRL** Rebuffi et al. (2017) we store means of classes' features for each ICs to use with NMC for prediction at the inference time. In all methods that leverage replay, we use the same batch of data for all the ICs and for the final classifer.

## 4  CONTINUAL EARLY-EXIT NETWORKS: AN ANALYSIS

In this section, we will provide an analysis of continually learned early-exit networks. We aim to shed light on the main problem of continual learning, namely *catastrophic forgetting* and one of the main network characteristics which serves as a motivation for early-exit networks called *overthinking*. Finally, based on the analysis, we propose an improved dynamic inference for early-exit networks in CL.

### 4.1  OVERTHINKING IN CONTINUAL LEARNING

*Overthinking* has been extensively studied for early-exit networks in the seminal work of Kaya et al. (2019). The concept of overthinking highlights the fact that inference-time speed-up does not directly need to result in a performance drop. To better understand the potential of early-exit networks for continually learned networks, we here study *overthinking* defined as the difference between the oracle performance and the actual performance. A sample is considered correctly classified by the oracle if there exist a single internal classifier which predicts the correct label.

We examine overthinking in early-exit networks trained with several exemplar-based and memory-free continual learning methods, including FT, FT+E, LwF, BiC, iCaRL and LUCIR. In Figure 2, we compare the performance of those methods with the network trained in a standard, joint setting, and find that continually trained early-exit networks are more prone to overthinking. This suggests that using early-exit networks might be more beneficial in continual scenario.

### 4.2  ANALYSIS OF CONTINUALLY LEARNED EARLY-EXIT NETWORKS

For this analysis, we perform CIL with early-exit network on a 10-split CIFAR100. We evaluate the individual accuracy of each IC in isolation after the full training and report the results in Figure 3. In Figure 3(a), we show the results for various continual learning algorithms. The results also include joint training (JT), a setting typically used in the existing literature on early-exit networks (Kaya et al., 2019; Matsubara et al., 2022), where the performance of the network goes up when moving towards the later classifiers. Interestingly, the results for fine-tuning (FT) show a different behavior,

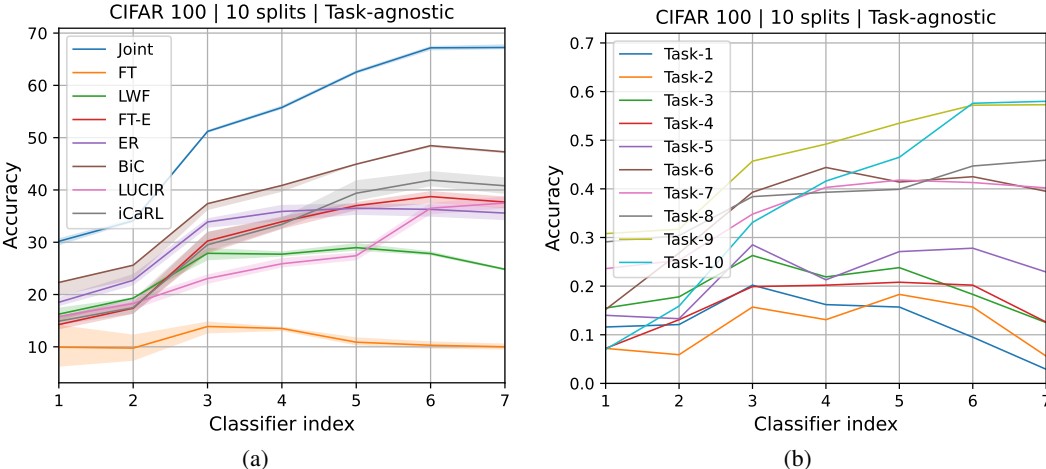

(a) (b)

Figure 3: CIL results on CIFAR100 (10-split) (a) Comparision of individual accuracy of each IC for models trained with various CL methods. Compared to standard joint training, CL methods suffer from less performance drop when moving to the early-exit framework. (b) Individual ICs performance on data from different tasks. We evaluate the model continually trained with LwF after the last task. Early ICs perform better than the later ones on the data from older tasks, indicating the potential of early-exit networks in continual learning.

where better performance is actually achieved by the earlier ICs. We hypothesize that this is caused by the fact that lower ICs suffer from less forgetting than the later ones and therefore obtain better overall performance when applied to data from all tasks. The continual learning methods perform with the accuracy between the FT and JT. The performance of exemplar-free method LwF at IC3 is similar to the performance at the final classifier (IC-7), while using only 47% FLOPs. Exemplar-based methods such as ER have a relatively straight performance curve up until IC-3. The best performing CL method, BiC, shows a clear drop-off – albeit slower than for JT – in performance when moving to lower ICs.

To better understand the performance curves observed in Figure 3(a), we perform an additional analysis where we plot the performance of a single method (LwF) on the separated data from the various tasks and show the results in Figure 3(b). When considering the curves for LwF, we see that for the last task data (task 10) the curve is similar as the JT curve (in Figure 3(a)). However, for previous tasks data, the curves change, and better performance is obtained at the earlier ICs (for example, consider the first task, where IC3 obtains 0.20 accuracy, while the final classifier almost totally forgets with 0.03 accuracy). This confirms our hypothesis that early ICs suffer from less catastrophic forgetting than later ICs. We evaluate several other CL methods and observe that this phenomenon is less evident for the exemplar-based methods, as expected based on the results from Figure 3(a). The results for the additional CL methods are provided in appendix A.2.

In conclusion, catastrophic forgetting manifests less in earlier layers of the network; this effect can be leveraged to reduce catastrophic forgetting in the continual learning scenario. Previous research already observed that continual learning mainly results in changes in the later layers of the network Liu et al. (2020); Zhao et al. (2023). Recently, Masarczyk et al. (2023) also showed that deep networks trained for image classification split into parts that build their representations differently. However, these works did not link their observations with any technique that could exploit them, like early-exits.

## 4.3 TASK-AWARE DYNAMIC INFERENCE

In Figure 5, we examine the average confidence of the correct predictions made by the early-exit network and notice that the average confidence for the latest task data is significantly higher than the confidence for data from previous tasks. During dynamic inference in early-exit networks, the network decides on the early-exit by comparing the confidence of its prediction with an exit threshold $\tau$. The results in Figure 5 show that it is significantly more likely to exit for later task data, and will probably only exit at the latest ICs for early task data. This is especially detrimental since we have seen in the previous sections that early ICs often yield higher performance for earlier task data.

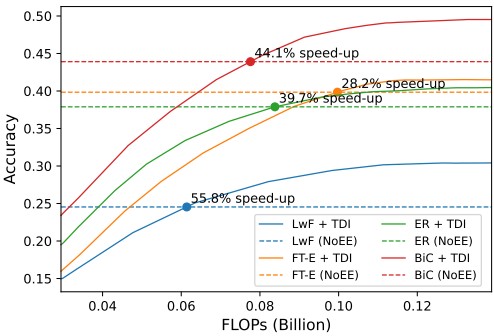 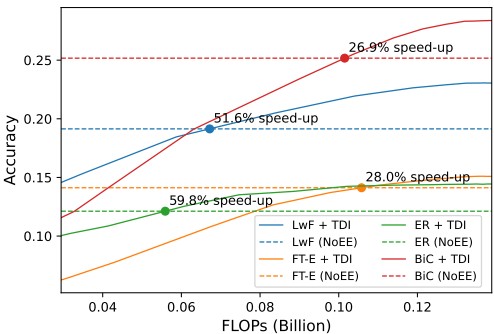

(a) 10-split setting on CIFAR 100 dataset.     (b) 10-split setting on TinyImageNet dataset.

Figure 4: Comparison between selection of methods with and without early-exit networks. The results here show that adapting early-exit framework can significantly improve the network performance in class-incremental learning. For all methods, we indicate the inference speed-up that can be obtained while maintaining the performance of the original algorithm.

To balance this bias, we propose *Task-aware Dynamic Inference (TDI)*, where we set different exit thresholds for different tasks:

$$\tau_t = \tau + (T - t) \cdot \alpha, \qquad (2)$$

where $t$ is the task index, $T$ is the total number of tasks and $\alpha$ is a hyperparameter. When multiple classes are above the threshold, the class which surpasses its threshold $\tau_t$ most is selected. This strategy leads to improved overall accuracy of the network.

The proposed task-aware dynamic inference is related to the task-recency bias, which has been studied extensively in continual learning (Wu et al., 2019; Ahn et al., 2021; Masana et al., 2022). It refers to the fact that the network predictions for the latest task are the most confident and accurate. We argue that this bias is especially detrimental within the context of early-exit network because it can lead to a significant amount of overthinking for early task data. In our work, we apply the same thresholding strategy for all the ICs, and we set $\alpha = 0.05$ in all our experiments. However, designing a strategy that takes into account the differences in confidence of the various ICs could potentially improve the results further, and we consider this as an promising future research direction.

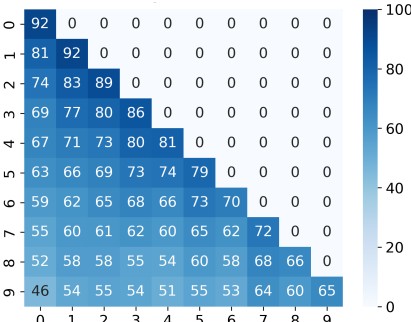

Figure 5: Average magnitude of confidence of the **correct** predictions of the final classifier for each task data on 10 splits of CIFAR100 over the course of training with LwF. The network is more confident about the predictions on the recently encountered data, which should be taken into account when performing dynamic inference in the early-exit network.

## 5  EXPERIMENTS

**Experimental setting.**  We conduct class-incremental continual learning experiments using the CIFAR100 and TinyImageNet datasets (task-incremental results are presented in the Appendix A.3). To assess the performance of different continual learning methods across varying lengths of task sequences, we partition the datasets into 5 or 10 disjoint classification tasks. Specifically, for the CIFAR100 dataset, this involves 5 tasks with 20 classes per task or 10 tasks with 10 classes per task, while for the TinyImageNet dataset, we split the data into 5 tasks with 40 classes per task or 10 tasks with 20 classes per task. We utilize the ResNet32 architecture, a standard network adopted in prior continual learning studies, and customize it with 6 early-exit internal classifiers following (Kaya et al., 2019). Refer to Table 5 in the Appendix A.4 for specific details regarding the early-exit ResNet32 architecture. We train the model for 200 epochs for both 5-split and 10-split settings. The learning rate is initialized to 0.1 and is decayed by a rate of 0.1 at the 60th, 120th, and 160th epoch.

| Method | 5-split | | | | | 10-split | | | | |
|---|---|---|---|---|---|---|---|---|---|---|
| | Speed-up ↑ | | | | | Speed-up ↑ | | | | |
| | **NoEE** | $\sim 0\%$ | $\sim 25\%$ | $\sim 50\%$ | $\sim 75\%$ | **NoEE** | $\sim 0\%$ | $\sim 25\%$ | $\sim 50\%$ | $\sim 75\%$ |
| JT | **68.41** | 66.76 | 65.94 | 55.24 | 33.28 | **68.41** | 67.10 | 66.29 | 55.91 | 33.88 |
| FT | 18.04 | 25.34 | 25.52 | 23.45 | 15.25 | 9.89 | **14.74** | 14.39 | 13.24 | 10.56 |
| +*TDI* | - | **25.61** | 25.37 | 22.73 | 14.73 | - | 13.98 | 13.97 | 12.66 | 9.96 |
| LwF | 39.22 | 43.61 | 43.05 | 37.98 | 25.18 | 25.03 | 29.07 | 28.67 | 25.90 | 17.98 |
| +*TDI* | - | **44.96** | 44.11 | 38.43 | 25.01 | - | **30.81** | 30.21 | 26.53 | 17.93 |
| FT-E | 40.75 | 41.02 | 39.38 | 31.23 | 17.28 | 38.35 | 38.11 | 35.66 | 29.46 | 16.6 |
| +*TDI* | - | **42.90** | 41.32 | 32.68 | 17.53 | - | **40.60** | 39.64 | 32.11 | 17.18 |
| ER | 38.03 | 40.21 | 40.44 | 37.47 | 26.8 | 36.99 | 38.32 | 37.86 | 33.64 | 21.86 |
| +*TDI* | - | **42.38** | 42.12 | 38.75 | 27.25 | - | **40.46** | 39.82 | 35.45 | 22.49 |
| BiC | 52.76 | 51.66 | 50.15 | 40.51 | 24.62 | 46.94 | 47.87 | 46.63 | 39.05 | 24.90 |
| +*TDI* | - | **52.94** | 51.68 | 41.75 | 24.85 | - | **49.03** | 47.61 | 40.27 | 25.25 |
| LUCIR | 44.56 | 43.48 | 37.24 | 27.95 | 20.04 | **39.53** | 38.14 | 33.08 | 24.97 | 17.03 |
| +*TDI* | - | **44.97** | 39.77 | 30.47 | 20.47 | - | 38.45 | 34.35 | 27.06 | 17.36 |
| iCaRL | **44.94** | 41.77 | 39.82 | 31.26 | 17.45 | 41.23 | 38.80 | 37.00 | 29.63 | 17.13 |
| +*TDI* | - | 44.30 | 42.25 | 32.81 | 17.79 | - | **42.51** | 40.88 | 32.67 | 17.83 |

Table 1: TAG final accuracy on CIFAR100 obtained with different amounts of compute. We evaluate CL methods extended with early-exit using early-exit inference as well as our proposed method of task-aware dynamic inference (TDI). For reference, we include the results for a standard, non-early-exit network in the same CL setting (NoEE). We include the upper bound of the model trained on a joint dataset (JT). Early-exit methods exhibit good compute-accuracy properties, and are often able to perform on par with methods applied to a standard network using significantly smaller compute and applying TDI often allows the early-exit method to outperform the standard counterpart.

We use SGD optimizer with a batch size of 128 for all methods (note that ER uses half of one batch for new data and half for the old ones from memory). For the CL methods with exemplars, we use a fixed memory of 2000 exemplars selected with herding (Rebuffi et al., 2017).

We investigate the compute-accuracy trade-off of early-exit CL methods described in Section 3.3. Next to fine-tuning(FT), we compare the exemplar-free method LwF, and five exemplar-based methods: finetuning with exemplars (FT-E), Experience Replay (ER), BiC, LUCIR and iCaRL. We examine the performance at the different compute thresholds and provide the results for the inference at different *speedups*. We measure the compute cost of inference with the early-exit networks using FLOPs, and calculate speedup as $1 - \frac{C_{EE}}{C_F}$, where $C_{EE}$ and $C_F$ are the cost of inference when exiting at the $i$-th IC and at the full network *without the ICs*. We also provide the reference results for a static inference with a standard network trained without early exits. For the upper bound, we also provide the results for the model trained on the joint dataset from all tasks jointly.

**Main results.** As demonstrated in Table 1 and Table 2, exemplar-free CL method LwF with early-exit exhibits superior performance compared to the NoEE counterparts on both the CIFAR 100 and TinyImageNet datasets for class incremental learning. Additionally, our proposed task-aware dynamic inference (TDI) algorithm significantly enhances the performance of LwF with negligible additional costs. Most notable, LwF with early exit can be employed with over 50% speed-up on the 10-split tasks at equal performance as standard LwF. At full computational capacity, the early-exit LwF version leads around 5% performance gain compared to its standard version. This shows that the early-exit strategy can be applied to improve continual learning; the data of the first tasks, which suffer especially from catastrophic forgetting in the last ICs, will be rooted to the early ICs, resulting in the dual benefit of early-exit networks by reducing forgetting while accelerating inference.

Next we look into the results for the exemplar-based methods. In the 5-split setting on the CIFAR 100 and TinyImageNet datasets, most CL methods with exemplars (except FT-E and ER) exhibit lower performance than their NoEE counterparts when operated at full capacity. However, with the incorporation of our proposed TDI, most surpass their NoEE counterparts (except iCaRL, which demonstrates a marginal 0.64% reduction compared to its NoEE counterpart). For the 10-split setting, aided by our proposed TDI, nearly all CL methods with early-exit networks outperform their

| Method | 5-split | | | | | 10-split | | | | |
|--------|---------|---|---|---|---|----------|---|---|---|---|
| | Speed-up ↑ | | | | | Speed-up ↑ | | | | |
| | **NoEE** | $\sim 0\%$ | $\sim 25\%$ | $\sim 50\%$ | $\sim 75\%$ | **NoEE** | $\sim 0\%$ | $\sim 25\%$ | $\sim 50\%$ | $\sim 75\%$ |
| JT | 45.88 | **46.79** | 43.17 | 32.34 | 20.94 | **45.88** | 44.91 | 41.89 | 32.89 | 23.74 |
| FT | 15.06 | **16.75** | 15.00 | 11.80 | 7.99 | 7.7 | **8.57** | 7.83 | 6.37 | 3.95 |
| +*TDI* | - | 15.63 | 14.19 | 10.86 | 7.58 | - | 6.46 | 6.09 | 5.06 | 3.47 |
| LwF | 31.73 | 34.53 | 32.09 | 25.84 | 18.11 | 19.14 | 24.35 | 23.16 | 19.92 | 15.31 |
| +*TDI* | - | **35.31** | 32.59 | 25.99 | 18.08 | - | **24.84** | 23.46 | 19.93 | 15.15 |
| ER | 15.55 | 17.16 | 16.92 | 15.23 | 13.03 | 12.12 | 13.64 | 13.55 | 12.55 | 10.31 |
| +*TDI* | - | **18.15** | 17.55 | 15.71 | 13.12 | - | 15.33 | 14.94 | 13.58 | 10.58 |
| FT-E | 17.02 | 17.66 | 16.20 | 12.88 | 9.41 | 14.12 | 13.56 | 12.64 | 9.97 | 6.83 |
| +*TDI* | - | **18.53** | 16.88 | 13.55 | 9.52 | - | **15.56** | 14.55 | 11.51 | 7.07 |
| BiC | 30.40 | 29.61 | 27.20 | 21.27 | 14.59 | 25.17 | 25.95 | 23.44 | 18.05 | 11.81 |
| +*TDI* | - | **30.96** | 28.47 | 22.25 | 14.75 | - | **27.16** | 24.65 | 19.27 | 12.09 |
| LUCIR | 23.77 | 22.66 | 19.07 | 15.39 | 12.20 | **21.47** | 18.20 | 15.54 | 12.53 | 9.71 |
| +*TDI* | - | **24.59** | 21.27 | 16.88 | 12.41 | - | 20.47 | 18.12 | 14.19 | 9.84 |
| iCaRL | **20.64** | 18.96 | 17.10 | 13.00 | 8.87 | 12.31 | 14.68 | 13.31 | 10.20 | 6.79 |
| +*TDI* | - | 20.41 | 18.15 | 13.58 | 8.92 | - | **17.98** | 15.98 | 11.49 | 6.70 |

Table 2: TAG final accuracy on TinyImageNet obtained with different amounts of compute. Our method of TDI provides a good cost-accuracy characteristic and often even allows the early exit model to outperform the standard one. Refer to Table 1 for the details of this table.

corresponding NoEE's ones on both the CIFAR 100 and TinyImageNet datasets (excluding LUCIR, which shows a $\sim 1\%$ decrease compared to its NoEE counterpart). Notable is the gain for the best performing method BiC, where especially for the more challenging 10-split setting the proposed TDI improvement leads to considerable performance gains.

In these experiments, we have set $\alpha = 0.05$ for all methods. A strategy which can find a method specific $\alpha$ (or by using a validation set) would surely allow for further improvements. We have included more results of the dependence on the $\alpha$ parameter for some methods in Appendix A.4.

The accuracy-efficiency comparison with early-exit network and NoEE is shown in Figure 4. It shows that significant speed-ups can be obtained while maintaining the same performance as the original algorithm. Overall, the experimental results underscore the effectiveness of our proposed task-aware dynamic inference approach, which notably enhances the performance of various CL methods during the inference phase when utilizing early-exit networks in continual learning.

## 6  Conclusion

We are the first to explore the continual learning of early-exit networks, which involves two critical challenges in practical applications: efficient inference on resource-limited devices, and continual learning with sequential data streams. We conduct an in-depth analysis of the performance of various early-exit classifiers in continual learning, demonstrating that they not only decrease inference time but also mitigate forgetting of previously encountered data. Additionally, we introduce a task-aware dynamic inference algorithm aimed at mitigating the negative impact of recency bias during inference in early-exit networks. The experimental results on CIFAR100 and TinyImageNet datasets validate the effectiveness of our proposed algorithm.

**Limitations.** While we are the pioneers in the exploration of continual early-exit network learning, there is still ample room for further investigation. For instance, conducting an in-depth analysis of the performance of continual early-exit networks with more advanced CL methods, studying the early-exit network in an online continual learning setting, and delving into various types of dynamic networks in continual learning. In addition, our proposed task-aware dynamic inference algorithm is simple and effective, it is conceivable that more sophisticated and advanced methods may emerge in the future.

**Reproducibility.** For easy reproducibility, we use the FACIL framework. Our source code is available for review in: `https://anonymous.4open.science/r/ContinualEE` and will be publicly available on github upon acceptance. We use standard hyperparameters for the used continual learning methods following (Masana et al., 2022). The only hyperparameter of our method is $\alpha$ which is set to constant for all methods, a further analysis of this parameter for some methods is included in Appendix A.4.

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

# A    APPENDICES

## A.1    CONTINUAL EARLY-EXIT NETWORKS WITH FINETUNING.

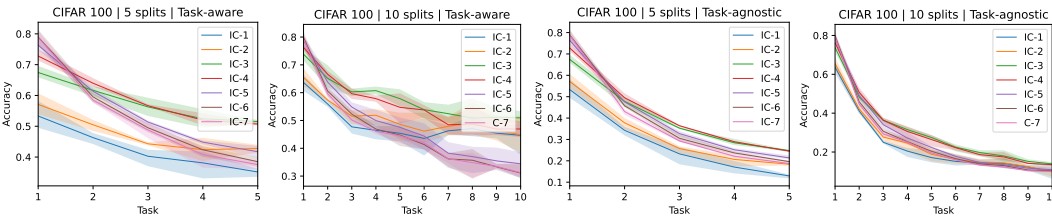

Figure 6: Finetuning without exemplars on CIFAR100 dataset.

In Figure 6, we present the performance of each classifier in a continually learned early-exit network using FT method at each task phase. In both task-incremental and class-incremental learning scenarios, the later classifiers exhibit more forgetting compared to the earlier ones, particularly noticeable in the last three classifiers.

## A.2    RESULTS FOR MORE CL METHODS WITH EARLY-EXIT NETWORKS.

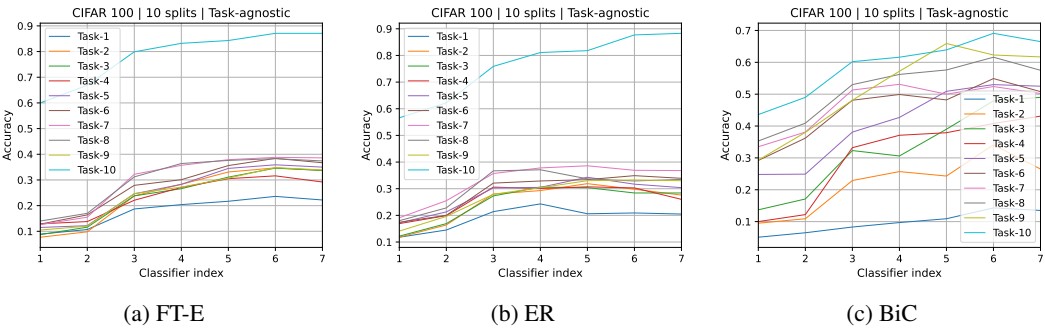

(a) FT-E             (b) ER             (c) BiC

Figure 7: CIL results of exemplar-based methods on CIFAR 100 (10-split).

Unlike the more conspicuous pattern shown by the exemplar-free method LwF, where early ICs perform significantly better for early task data, exemplar-based methods weaken a bit this phenomenon as shown in Figure 7. In these methods, early ICs maintain slightly higher or similar performance levels compared to later ICs for previous task data. Additionally, for FT-E and ER, there is a noticeable performance gap between new and old task data, whereas this distinction is absent in BiC due to the bias removal facilitated by its prediction rectification layer.

## A.3    TASK-INCREMENTAL RESULTS

For the two exemplar-free methods considered in our experiments, when combined with an early-exit network for task incremental learning, both of them demonstrate significant performance improvements compared to training the standard network (only with the final classifier, referred to as 'NoEE' in the paper). Specifically, FT exhibits better performance than its corresponding NoEE version, even with a 75% increase in inference speed on both CIFAR100 and TinyImageNet datasets (refer to Table 3 and Table 4). Furthermore, achieving the best performance for FT with early-exit network requires only 50% of the inference computation cost, showing its efficiency for practical resource-limited applications. In the case of LwF, a 25% and 50% speed-up, compared to its corresponding NoEE, can be attained while maintaining the same level of performance in the 5-split and 10-split settings, respectively. As for the methods that store exemplars in memory, their performance at full inference capacity is similar to their corresponding NoEE's ones, except for FT-E, which shows similar performance with a 25% and 50% speed-up for the 5-split and 10-split settings on the CIFAR 100 dataset, and more than 25% speed-up on the TinyImageNet 10-split setting.

| Method | 5-split | | | | | 10-split | | | | |
|---|---|---|---|---|---|---|---|---|---|---|
| | **Speed-up ↑** | | | | | **Speed-up ↑** | | | | |
| | **NoEE** | $\sim 0\%$ | $\sim 25\%$ | $\sim 50\%$ | $\sim 75\%$ | **NoEE** | $\sim 0\%$ | $\sim 25\%$ | $\sim 50\%$ | $\sim 75\%$ |
| JT | **85.30** | 83.64 | 83.56 | 78.73 | 56.59 | **90.53** | 89.30 | 89.30 | 87.06 | 68.26 |
| FT | 36.12 | **51.73** | 52.26 | 52.08 | 39.57 | 32.34 | **50.91** | 51.40 | 52.05 | 47.29 |
| LwF | 74.18 | **75.73** | 75.58 | 71.82 | 54.88 | 72.53 | **78.12** | 78.04 | 76.18 | 63.69 |
| FT-E | 69.40 | **71.71** | 71.51 | 66.65 | 48.35 | 74.98 | **77.30** | 77.24 | 74.72 | 58.91 |
| ER | **66.61** | 65.01 | 64.87 | 60.91 | 46.82 | 74.05 | **74.09** | 74.07 | 72.07 | 58.57 |
| BiC | **78.31** | 77.91 | 77.79 | 73.19 | 54.02 | 81.72 | **81.77** | 81.73 | 79.29 | 64.54 |
| LUCIR | **76.07** | 74.57 | 73.63 | 64.20 | 45.91 | **78.73** | 76.94 | 76.42 | 70.52 | 54.71 |
| iCaRL | **78.11** | 77.35 | 77.18 | 72.36 | 54.60 | 81.65 | **81.88** | 81.79 | 79.49 | 65.30 |

Table 3: TAW results on CIFAR100.

| Method | 5-split | | | | | 10-split | | | | |
|---|---|---|---|---|---|---|---|---|---|---|
| | **Speed-up ↑** | | | | | **Speed-up ↑** | | | | |
| | **NoEE** | $\sim 0\%$ | $\sim 25\%$ | $\sim 50\%$ | $\sim 75\%$ | **NoEE** | $\sim 0\%$ | $\sim 25\%$ | $\sim 50\%$ | $\sim 75\%$ |
| JT | 65.94 | **66.36** | 64.96 | 54.99 | 39.16 | **74.19** | 73.04 | 72.58 | 66.50 | 52.75 |
| FT | 27.69 | **36.58** | 36.73 | 33.77 | 26.03 | 23.85 | **34.82** | 35.25 | 34.09 | 26.82 |
| LwF | 57.85 | **60.32** | 58.94 | 50.96 | 38.49 | 51.06 | **61.87** | 61.38 | 57.69 | 47.44 |
| ER | 38.47 | **39.23** | 38.39 | 34.72 | 29.08 | 43.92 | **46.25** | 45.79 | 43.36 | 36.67 |
| FT-E | 44.06 | **44.81** | 43.39 | 37.92 | 29.48 | 46.91 | **48.92** | 48.25 | 44.31 | 35.89 |
| BiC | **58.27** | 58.22 | 57.15 | 50.15 | 38.99 | 59.36 | **61.11** | 60.41 | 54.33 | 42.34 |
| LUCIR | 60.56 | 57.25 | 53.96 | 43.99 | 32.05 | **62.20** | 60.65 | 57.97 | 48.80 | 37.24 |
| iCaRL | **57.93** | 57.69 | 56.25 | 49.14 | 38.54 | 56.46 | **59.80** | 58.98 | 53.25 | 42.80 |

Table 4: TAW results on TinyImageNet200.

One notable phenomenon is that LwF achieves comparable performance to the exemplar-based methods on the CIFAR 100 dataset at different levels of speed-up, and even outperforms them on the TinyImageNet dataset.

A.4   HYPERPARAMETERS OF TDI.

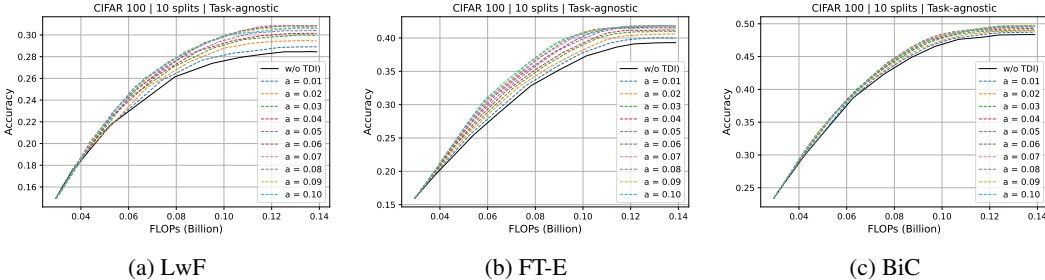

(a) LwF              (b) FT-E              (c) BiC

Figure 8: Results of TDI when using different hyperparameter setting for $\alpha$.

As depicted in Figure 8, we illustrate the impact of the hyperparameter of $\alpha$ in our proposed TDI for various CL methods. We experimented with different values for $\alpha$ within the range of $[0.01, 0.1]$, all of which show performance improvement. Thus, we just opted for a uniform value of $\alpha = 0.05$ for all methods considered in this paper, which has already yielded enhanced performance. Nevertheless, more sophisticated strategies can be explored in future work.

## A.5 NETWORK ARCHITECTURE

Table 5 and Table 6 display the details of our implemented early-exit ResNet 32 for CIFAR 100 and TinyImageNet datasets, respectively. It's worth noting that there is a difference in resolution between CIFAR 100 and TinyImageNet. Consequently, we have adapted the size of the fully connected layers for the internal classifiers accordingly.

| LAYER | DOWNSAMPLE | OUTPUT SHAPE |
|---|---|---|
| Input | - | $32 \times 32 \times 3$ |
| Conv1×1 | - | $32 \times 32 \times 16$ |
| BN | - | $32 \times 32 \times 16$ |
| ReLU | - | $32 \times 32 \times 16$ |
| ResBlk | - | $32 \times 32 \times 16$ |
| ResBlk | - | $32 \times 32 \times 16$ |
| ResBlk | - | $32 \times 32 \times 16$ |
| FR (early exit) | MixPool | $4 \times 4 \times 16$ |
| FC (early exit) | - | NumOfClasses |
| ResBlk | - | $32 \times 32 \times 16$ |
| ResBlk | - | $32 \times 32 \times 16$ |
| FR (early exit) | MixPool | $4 \times 4 \times 16$ |
| FC (early exit) | - | NumOfClasses |
| ResBlk | stride=2 | $16 \times 16 \times 32$ |
| ResBlk | - | $16 \times 16 \times 32$ |
| FR (early exit) | MixPool | $4 \times 4 \times 32$ |
| FC (early exit) | - | NumOfClasses |
| ResBlk | - | $16 \times 16 \times 32$ |
| ResBlk | - | $16 \times 16 \times 32$ |
| FR (early exit) | MixPool | $4 \times 4 \times 32$ |
| FC (early exit) | - | NumOfClasses |
| ResBlk | stride=2 | $8 \times 8 \times 64$ |
| ResBlk | - | $8 \times 8 \times 64$ |
| FR (early exit) | MixPool | $4 \times 4 \times 64$ |
| FC (early exit) | - | NumOfClasses |
| ResBlk | - | $8 \times 8 \times 64$ |
| ResBlk | - | $8 \times 8 \times 64$ |
| FR (early exit) | MixPool | $4 \times 4 \times 64$ |
| FC (early exit) | - | NumOfClasses |
| ResBlk | - | $8 \times 8 \times 64$ |
| ResBlk | - | $8 \times 8 \times 64$ |
| - | AvgPool | $1 \times 1 \times 64$ |
| FC (Final) | - | NumOfClasses |

Table 5: Early-exit ResNet32 for the CIFAR100 dataset.

| LAYER | DOWNSAMPLE | OUTPUT SHAPE |
|---|---|---|
| Input | - | $64 \times 64 \times 3$ |
| Conv1×1 | - | $64 \times 64 \times 16$ |
| BN | - | $64 \times 64 \times 16$ |
| ReLU | - | $64 \times 64 \times 16$ |
| ResBlk | - | $64 \times 64 \times 16$ |
| ResBlk | - | $64 \times 64 \times 16$ |
| ResBlk | - | $64 \times 64 \times 16$ |
| FR (early exit) | MixPool | $8 \times 8 \times 16$ |
| FC (early exit) | - | NumOfClasses |
| ResBlk | - | $64 \times 64 \times 16$ |
| ResBlk | - | $64 \times 64 \times 16$ |
| FR (early exit) | MixPool | $8 \times 8 \times 16$ |
| FC (early exit) | - | NumOfClasses |
| ResBlk | stride=2 | $32 \times 32 \times 32$ |
| ResBlk | - | $32 \times 32 \times 32$ |
| FR (early exit) | MixPool | $8 \times 8 \times 32$ |
| FC (early exit) | - | NumOfClasses |
| ResBlk | - | $32 \times 32 \times 32$ |
| ResBlk | - | $32 \times 32 \times 32$ |
| FR (early exit) | MixPool | $8 \times 8 \times 32$ |
| FC (early exit) | - | NumOfClasses |
| ResBlk | stride=2 | $16 \times 16 \times 64$ |
| ResBlk | - | $16 \times 16 \times 64$ |
| FR (early exit) | MixPool | $8 \times 8 \times 64$ |
| FC (early exit) | - | NumOfClasses |
| ResBlk | - | $16 \times 16 \times 64$ |
| ResBlk | - | $16 \times 16 \times 64$ |
| FR (early exit) | MixPool | $8 \times 8 \times 64$ |
| FC (early exit) | - | NumOfClasses |
| ResBlk | - | $16 \times 16 \times 64$ |
| ResBlk | - | $16 \times 16 \times 64$ |
| - | AvgPool | $8 \times 8 \times 64$ |
| FC (Final) | - | NumOfClasses |

Table 6: Early-exit ResNet32 for the TinyImageNet dataset.

