# OpenReview forum: "Accelerated Inference and Reduced Forgetting: The Dual Benefits of Early-Exit Networks in Continual Learning"
_ICLR.cc/2024/Conference — ICLR 2024 Conference Withdrawn Submission_

### Official Review · Reviewer_AuwN · 2023-10-26

**Soundness:** 3 good
**Presentation:** 3 good
**Contribution:** 3 good
**Rating:** 6
**Confidence:** 4

**Summary:**

This paper studies dynamic early exiting in the continual learning setting. Analytic experiments show that early exiting may, to some extent, mitigate the catastrophic forgetting problem. Furthermore, the authors observed that the traditional early-exiting strategy is not suitable for continual learning, since a multi-exit model tends to have higher confidence for new tasks. Based on this, a threshold calibration trick is proposed, which improves the performance of early-exiting models in continual learning.

**Strengths:**

1. The studied problem is important and interesting;

2. Extensive experiments are conducted on multiple continual learning methods, which consistently demonstrate the claims of authors and the effectiveness of the proposed method;

3. The proposed threshold calibration is simple yet effective.

**Weaknesses:**

My biggest concern mainly involves three parts:

1. **Insufficient early-exiting models are evaluated**. Only a small Shallow-Deep ResNet-32 is examined. However, there are various superior dynamic early-exiting models in computer vision, such as MSDNet [1], RANet [2], CF-ViT [3] and Dyn-Perceiver [4]. It is strongly suggested that the authors examine their claims on these more SOTA models.

2. **Only toy datasets are used**. As we all know, Cifar100 and Tiny-ImageNet are both small-scale toy datasets. Therefore, It is kindly recommended to conduct experiments on the standard ImageNet dataset.

3. **The proposed TAD is not studied in depth**. The linear-decay form lacks theoretical guarantee or empirical comparison with other forms.

[1] Huang et al, Multi-Scale Dense Networks for Resource Efficient Image Classification.

[2] Yang et al, Resolution Adaptive Networks for Efficient Inference.

[3] Chen et al, CF-ViT: A General Coarse-to-Fine Method for Vision Transformer.

[4] Han et al, Dynamic Perceiver for Efficient Visual Recognition.

**Questions:**

See weaknesses.

---

### Official Review · Reviewer_Xsg7 · 2023-10-30

**Soundness:** 2 fair
**Presentation:** 2 fair
**Contribution:** 2 fair
**Rating:** 3
**Confidence:** 4

**Summary:**

This paper studies applying early exit image classifiers (ResNets) in a continual learning (CL) setup. The paper analyzes the accuracy and softmax-based confidence measures across different exit gates. The findings (e.g. figure 3) indicate that while for more recently observed tasks the accuracy increases throughout the layers of the model (as expected), for older tasks the performance stays roughly stationary. The authors conclude that older tasks can benefit more from early exiting, and that different tasks can use different confidence thresholds for exiting.

The authors propose to adjust the exiting confidence threshold according per task (called TDI). Experiments are performed on CIFAR100 and TinyImageNet datasets in a continual learning setup with multiple different CL methods. TDI provide better accuracy for matching speedup, though as expected all methods are far behind the non-CL setting that trains on all tasks jointly (JT).

**Strengths:**

1. Studying early exits in the context of continual learning is interesting.
2. The papers presents a detailed analysis and experiments with multiple different methods.
3. The proposed threshold adapting method is very simple, yet the experiments found it to be effective.

**Weaknesses:**

1. The proposed TDI method requires knowing the total number of tasks $T$ which is not realistic for continual learning setup, and the specific task index $t$ which doesn't aligned with the experimented task-agnostic learning (TAG).
2. Parts of the analysis are not convincing enough and the presented results don't always align with the claims in the text. For example, figure 5 only examines the confidence of the final layer, so it doesn't prove the claim in the text that " it is significantly more likely to exit for later task data". The confidence of earlier layers (and taking into account the exit policy) should be known for examining this. Also, the experiment in Figure 3 (b) should be repeated with different shuffling of the tasks to allow disentangling ordering of the tasks vs. task-specific features.
3. The main repeated claim that the observed pattern "can be leveraged to reduce catastrophic forgetting in the continual learning scenario" also doesn't sound accurate. On the best performing CL method (BiC), early exits reduce the accuracy (Tables 1,2). Perhaps claiming that early exit models are less affected by catastrophic forgetting would be more aligned with the findings?
4. Some details in the writing could be improved to flow better and help clarify the experiments. For example, sec 2.3 presents many CL methodologies but doesn't position them with the rest of the paper. sec 3.1 presents some settings (TIL, CIL, TAG, TAW), but the rest of the paper doesn't stick to them. The compute of FLOPs for speedup should be clarified, and what does it mean ~0% speedup in the tables? It's not entirely clear if comparing methods within the same column is fair or could there be differences in the actual speedup.

**Questions:**

please see questions and points in the weaknesses part above

---

### Official Review · Reviewer_cGA4 · 2023-10-31

**Soundness:** 3 good
**Presentation:** 4 excellent
**Contribution:** 2 fair
**Rating:** 3
**Confidence:** 4

**Summary:**

This submission examines the interplay of jointly applying early-exiting (EE) and continual learning (CL) on a CNN backbone. Initially, an analysis finds  some interesting properties, namely: i) continually learned EE-networks are more prone to overthinking. ii) the shallow classifiers on continually learned EE-networks are more resilient to catastrophic forgetting than the deeper ones and iii) the average confidence of the most recent task used in CL  is higher compared to previous tasks.

Driven by the third observation, the submission introduces a decaying confidence threshold on the exit-policy of each intermediate classifier (considering the prediction confidence to determine whether computation should continue or "exit-early"), applicable on task-aware CL where the last time a specific task was met can be tracked and considered to lower the confidence thresholds at runtime.

**Strengths:**

-The paper studies the intersection of two prominent and very important research fields in AI, namely Continual Learning and Dynamic Inference.

-The conducted analysis offers plenty of insights (summarised above), that can drive further research in the field.

-A wide variety of CL methods are adopted and comparatively tested in the proposed setting, offering insightful take-aways.

-The manuscript is well-written and easy-to-follow and features helpful illustrations of different concepts.

**Weaknesses:**

-Following the strong analysis part, the proposed solution only takes advantage of the 3rd insight (as listed above); and while the 1st can also be consider a  direct benefit of combining EE and CL; and the 2nd insight does not appear to be exploited at all. Additionally, the proposed TDI, is only applicable on a small subset of the examined cases, where the task-id at each CL-stage is available, and task classes are disjoint. Overall, although the paper offers several of useful insights from the preceding analysis, it lack a novel technical contribution that takes sufficient advantage of these findings.

-The breadth of the evaluation (as well as analysis in Sec.4) in terms of the EE-models adopted is severely limited to help guide conclusive results. In reference a single CNN-based EE model is used and evaluated on Cifar-100 (and in some cases on TinyImagenet). More robust experimental evaluation and analysis of key finding is required to validate their robustness, considering more complex EE architectures (e.g. MSDNet) as well as validate that the results hold for EE-based vision transformers.

Given the above, I believe this work takes a very promising approach, but is not yet ready for publication at ICLR; but strongly encourage the authors to continue their investigation in this field extending the current progress.

**Questions:**

1) How would the proposed method behave in the (more realistic) case were there is class overlap between different tasks. Can the proposed TDI method be adapted to accommodate this ?

2) Why was TinyImageNet used, in contrast to the full ImageNet-1K, which is a common baseline for early-exit models.

3) Do the results of the analysis (Sec.4) continue to stand for different EE-models.